# A preliminary study into internet related addictions among adults with dyslexia

**Suresh Kumar[1], Sophie Jackson [2]° \*, Dominic Petronzi [3]°**

1 Lazarus Centre Pte Ltd, Singapore, Singapore, 2 Department of Psychology, School of Social Sciences, Birmingham City University, Birmingham, United Kingdom, 3 School of Psychology, University of Derby, Derby, United Kingdom

☉ These authors contributed equally to this work.
\* Sophie.Jackson2@BCU.ac.uk

## Abstract

In recent decades, studies have investigated associations between learning disorders such as Autism Spectrum Disorder (ASD) and Attention Deficit Hyperactivity Disorder (ADHD), and the various types of internet addictions, ranging from general internet addiction (GIA) to specific internet addictions such as social media addiction (SMA) and internet gaming disorder (IGD). However, to date, no study has investigated such internet addictions among persons with dyslexia. The present study aimed to investigate whether differences exist between adults with dyslexia and controls in terms of GIA, SMA and IGD. A total of 141 adults with dyslexia and 150 controls (all UK based) were recruited. Controlling for age, gender, marital status, employment, and income levels, it was found that adults with dyslexia had higher levels of GIA and IGD compared to controls. However, these participants did not show any significant difference in terms of SMA. The results indicate that internet addictions may have a larger ambit for learning disorders beyond just ASD and ADHD and could be a hidden problem for these individuals.

## Introduction

The internet continues to be a popular platform for information seeking, education and entertainment, in addition to social interaction and online games. However, there are concerns over addictive usage among a minority of users, this includes those with learning disabilities [1]. Such an addiction has been defined as General Internet Addiction (GIA) and includes a preoccupation with internet activities at the expense of important daily activities such as schoolwork, occupation, relationships, and personal health [2]. These addictions can also be unique to social networking or social media (named Social Media Addiction; SMA; [3]), or exclusive to internet games, known as Internet Gaming Disorder (IGD; [4]). There is much literature suggesting that all these forms of addictions are high in those with learning disorders but notably, much of this literature has focused solely on those with Autism Spectrum Disorder (ASD) and Attention Deficit Hyperactivity Disorder (ADHD) [5] Indeed, to date, no study has investigated such internet addictions among persons with dyslexia, a condition characterised by deficits in word decoding, spelling, reading fluency and comprehension [6] and which accounts for 10–15% of the UK population [7].

**Data Availability Statement:** All relevant data are within the paper and its Supporting information files.

**Funding:** Our study was funded by a private funder (Mr Bobby Lim). The funder had no role in study

design, data collection and analysis, decision to publish, or preparation of the manuscript and none of the authors receive a salary from this funder.

**Competing interests:** The authors have declared that no competing interests exist.

It is likely that dyslexia might be associated with these types of addictions because a growing body of evidence suggests that individuals with learning disabilities are especially vulnerable to internet addictions compared to their typically developing peers. For instance, studies have found a significant association between General Internet Addiction (GIA) and both ASD [5] and ADHD [8]. Similarly, Internet Gaming Disorder (IGD) has also been associated with ASD [9] and ADHD [10]. In ASD this may be due to restricted and repetitive interests (a core symptom of ASD) leading to difficulties in disengaging from video games or time spent on the internet and therefore an addiction [11]. In addition, the low social demands and audio-visual and structural characteristics of the internet and games may further add to the appeal [12]. In ADHD, being bored easily and an aversion for delayed reward are two key symptoms and therefore the internet and gaming may be especially appealing to these individuals, and it provides a variety of activities, many with instant rewards [8]. Additionally, neurological research has found abnormal brain activities in both those with ASD and ADHD which lead to impaired inhibition and lack of self-control ability [8, 11]. Given that those with dyslexia also show impairments on a range of executive functions, including inhibition and self-control [12, 13], links to internet related addictions are likely.

Research also shows a significant link between Social Media Addiction (SMA) and ADHD [14], again perhaps because of the instant rewards social media can offer such as 'likes' from peers and other users. Yet the relationships between SMA and ASD is unclear; while one study [15] found that children with ASD (n = 202) spent less time on social media than their typically developing siblings (n = 179), another study found no difference in time spent on social media among adolescents with and without ASD (ASD n = 24, control n = 26) [16]. Meanwhile, another study found that the majority of adults with ASD used social media to connect with others [17] perhaps because they find social engagement through the written form more appealing and less challenging than engaging with peers orally such as face-to-face or over the phone, something that may not be the case for those with dyslexia. Yet, these contrasting findings are perhaps due to age differences and the fact that children, adolescents, and adults may use social media for different purposes.

Despite some contradictory findings regarding SMA and ASD, taken together these research studies clearly highlight a link between ASD and ADHD and internet-based addiction. In addition to the ones already discussed, another explanation for this link may be due to ASD and ADHD triggering mental health conditions which are in turn a risk factor for internet addictions. For instance, ASD has reportedly induced anxiety [18], which is an antecedent for internet-related addictions [18–21]. Similarly, children with ADHD present with anxiety, depression, and poor self-esteem [22–24]. As dyslexia also triggers similar mental health issues, such as anxiety and low self-esteem [25], a similar relationship may exist between dyslexia GIA, IGD and SMA.

The link between dyslexia and internet gaming seems likely. This is because online games typically do not involve writing and thus have fewer spelling demands. It is logical to suggest therefore that such an environment would be highly appealing to those with dyslexia. For instance, some studies aimed at using video games as interventions for those with dyslexia have demonstrated that action video games provide a rewarding experience that reinforces the engagement for users with dyslexia [26, 27]. However, if this leads to high prevalence of IGD in this population is something which has yet to be explored. Hence this was akey aim of the present study.

On the other hand, the link between SMA and dyslexia is harder to explain as there is some evidence that suggests barriers for usage of social media. For example, the spelling deficits and comprehension difficulties associated with dyslexia may make using social media extremely challenging. Indeed, a study on how students (n = 40) used a library information system

(without spelling support) showed that spelling deficits hampered those with dyslexia as compared to typically developing peers, with users with dyslexia spending more time searching compared to their peers [28]. Moreover, another study [29] reported that 48% of participants with dyslexia (n = 67) received significantly more peer negative feedback on their social media posts as compared to about 22% of controls (n = 404). They cited spelling as the main reason why writing was harder than reading on social media sites [30]. Similarly, comprehending or integrating information when presented in various formats is a common challenge for those with dyslexia and one that could create problems when using social media. In a study of tenth-grade Norwegians (n = 44), it was found that typically developing individuals outperformed participants with dyslexia on synthesizing information across different web pages [31]. Likewise, studies [e.g., 32] have shown that when information is presented in different formats (text, images, videos etc) on a page with use of cluttered spacing, variety of colours, multiple columns, and lengthy sentences without bullet points, which can be common on social media sites, this could be difficult for persons with dyslexia to follow [33, 34].

Nonetheless, while spelling deficits and information integration are major issues for those with dyslexia, anecdotal evidence suggests some do employ coping strategies when using the internet. One strategy for searching information is to use search engines (such as Google) because they provide query suggestions and are tolerant of spelling errors [35]. This type of strategy was reported in a qualitative study where participants with dyslexia talked positively about using Facebook and stated they coped with their spelling deficits by using external resources such as MS word and Google. Similarly, research with students has shown that despite struggling to integrate academic information across multiple sources as compared to their peers (n = 20), some undergraduates with dyslexia (n = 13) went online to look for videos (YouTube) instead of relying on their prescribed readings [36].

In summary, studies have shown that spelling deficits and information integration difficulties are perhaps barriers to using social media for those with dyslexia suggesting that those with dyslexia are not likely to be susceptible to SMA. Yet coping strategies may help mitigate the challenges and therefore research is needed to identify if those with dyslexia are susceptible to SMA. Therefore, the current study aimed to shed light on this. On balance, given that those with ASD are not susceptible to SMA—and because it is noted that spelling deficits and poor comprehension are life-long challenges and hence permanent aspects of life for those with dyslexia, we argue that it is likely that users with dyslexia would naturally avoid or at least have lower levels of SMA as compared to controls. This is because social media platforms such as Twitter or Facebook do not, in general, provide spell check functions that could assist the writer when drafting a post for public viewing and while some to attempt to use third party applications (e.g., Google Chrome, Microsoft Word) to check their spelling before posting the fear of spelling remains a major deterrent. Hence exploring whether this is the case will also be a key aim of this study.

Literature has suggested that some types of social demographics may be associated with various internet related addictions, specifically, age and gender. In typically developing populations, age has been shown to be negatively and significantly associated with GIA [37] and SMA [38] with younger individuals showing higher levels of addiction. However, findings are mixed for IGD [39, 40]. Age is also shown to be negatively and significantly related to these types of addiction in both ASD and ADHD populations [41–43].

As for gender, literature suggests that womanare more likely to show SMA as opposed to men [44], while men are more likely to have a GIA [45] and IGD [46]. As for ASD populations, these findings are shadowed with research showing that more men with ASD than woman play video-action games [47]. Given these links, it is important that work into these internet addictions controls for such demographic factors. Furthermore, given that dyslexia, and

indeed ASD and ADHD, are reported to be more prevalent in men [48], this further demonstrates the need for controlling for gender in research in this area.

## The present study

The present study aimed to investigate whether differences exist between a UK sample of participants with dyslexia and controls in terms of GIA, SMA and IGD. Despite much evidence showing links to these types of addiction and other forms of learning disability no research has explored these forms of addiction in relation to dyslexia. Such research is warranted because if significant links between dyslexia and problematic internet usage are identified, early detection and targeted interventions can be formulated to mitigate risks for such this group.

The following hypothesis were investigated. After controlling for age, gender, income levels, marital status and educational levels: Adults with dyslexia will have significantly higher levels of GIA as compared to controls without a dyslexia diagnosis (**Hypothesis 1**); adults with dyslexia will have significantly higher levels of IGD as compared to controls without a dyslexia diagnosis (**Hypothesis 2**), and adults with dyslexia will have significantly lower levels of SMA as compared to controls without a dyslexia diagnosis (**Hypothesis 3**).

## Method

### Design

The study utilised a quantitative between-subjects design and used a convenience sample of UK adults. The dependent variables were GIA, SMA and IGD. The independent variable was dyslexia (Level 1 = no dyslexia diagnosis, Level 2 = dyslexia diagnosis). The other fixed factors were gender, education level, marital status, and income levels. The covariate was age. Details regarding the definitions and scoring of the variables are provided in the materials subsection.

### Participants

Participants were recruited through Prolific (an online survey platform). In the first step, participants with dyslexia were recruited; the inclusion criteria were a formal dyslexia diagnosis and no other learning disorders and no active ill mental health. A total of 141 participants with dyslexia completed the survey. In the second step, controls were recruited; the inclusion criteria were no dyslexia diagnosis, no other learning disorders and no active ill mental health. A total of 150 controls completed the survey. All participants were located in the UK and aged 18 and above. The mean age of controls and participants with dyslexia diagnosis was 39.4 ($SD$ = 14.5) and 43.2 ($SD$ = 11.0) years old respectively. Participants were recruited between 22 and 25th February (2022) and were paid approximately £1 for their participation). See Table 1 for full demographics.

### Materials

GIA was measured by the Internet Addiction Test [(IAT; [49]). The IAT is based on the DSM-IV criterion for pathological gambling diagnosis. There are 20 questions (e.g., "*How often do you find that you stay on-line longer than you intended*?") with six options ranging from Does Not Apply (0) to Always (5). The total score ranges from 0 to 100, interpreted using the following cut-offs: severe (80 and above), moderate (50 to 79), mild (31 to 49) and no addiction or normal usage (0 to 30) [2]. An independent study reported Cronbach's alpha (α) of .90, test-retest reliability of .83 and convergent validity range of .62–.84 [50]. In the present study, α = .93 indicating excellent internal consistency.

**Table 1. Sociodemographic characteristics of participants.**

| | Control | | Dyslexia | | Full sample | |
|---|---|---|---|---|---|---|
| | n | % | n | % | n | % |
| **Gender** | | | | | | |
| Female | 36 | 24 | 64 | 45 | 100 | 34 |
| Male | 113 | 75 | 73 | 52 | 186 | 64 |
| Others | 1 | 1 | 4 | 3 | 5 | 2 |
| **Marital Status** | | | | | | |
| Married | 73 | 49 | 53 | 38 | 126 | 43 |
| Single | 70 | 47 | 72 | 51 | 142 | 49 |
| Divorced/Widow | 7 | 5 | 16 | 11 | 23 | 8 |
| **Income** | | | | | | |
| Above 62,400 | 29 | 19 | 30 | 21 | 59 | 20 |
| 64,200 to 29,900 | 74 | 49 | 58 | 41 | 132 | 45 |
| Below 13,800 | 32 | 21 | 30 | 21 | 62 | 21 |
| **Education** | | | | | | |
| Primary/Sec. | 21 | 14 | 22 | 16 | 43 | 15 |
| College/Diploma | 42 | 28 | 35 | 25 | 77 | 26 |
| Degree | 53 | 35 | 49 | 35 | 102 | 35 |
| Masters/PhD | 34 | 23 | 35 | 25 | 69 | 24 |
| **Employment** | | | | | | |
| Unemployed | 5 | 3 | 8 | 6 | 13 | 4 |
| Not working | 22 | 15 | 12 | 9 | 34 | 12 |
| Employed | 87 | 58 | 91 | 65 | 178 | 61 |
| Self-Employed | 15 | 10 | 17 | 12 | 32 | 11 |
| Studying | 21 | 14 | 13 | 9 | 34 | 12 |

Total sample is 291; Dyslexia diagnosis (141), Controls (150)

SMA was measured by the Bergen Social Media Addiction Scale (BSMAS; [38]). The scale is based on the six core components model (salience, mood, modification, tolerance, withdrawal conflict and relapse) proposed by Griffiths to assess social media addiction [21]. The BSMAS is a modified version of the Bergen Facebook Addiction Scale (BFAS; [51]); questions were modified by using the word "social media" instead of "Facebook". There are six questions (e.g., "*How often during the last year have you felt an urge to use social media more and more*?"). Participants rate all items on a 5-point Likert scale ranging from Very Rarely (1) to Very Often (5). The total score ranges from 6 to 30. Higher scores indicate higher levels of addiction. Scores above 24 may be indicative of severe addiction and above 18, moderate addiction [52]. The internal consistency of the present study compared favourably ($\alpha = .91$) with the original study ($\alpha = .88$; [34]).

IGD was measured by the Internet Gaming Disorder Scale, Short-Form 9 (IGDS-SF9; [4]). The measure includes 9 questions (e.g., "*Have you ever continued your gaming activity despite knowing it was causing problems between you and other people*?") rated on a five-point Likert scale, ranging from Never (1) to Very Often (5). The total score ranges from 9 to 45. A higher score indicates a higher likelihood of IGD. A score above 32 is indicative of pathological usage based on Qin [53] who suggested that such a score was adequate to distinguish disordered and non-disordered gamers. A recent study reported $\alpha = .91$ [54]. Again, the present study demonstrated strong internal reliability ($\alpha = .95$) in comparison to previous works.

## Procedure

Participants who signed up for the survey were given a link to Qualtrics where they read the participant information sheet before providing online written informed consent. Participants were guided to click the consent button to proceed to the online survey. They also agreed to the GDPR statement before generating a unique user code. Participants then completed the questions on internet addiction, social media addiction, and internet gaming disorder IGD before providing demographic information (e.g., age, gender, and household income). Lastly, they reaffirmed their consent and viewed the project debrief information. Ethical approval was granted by the University of Derby research ethics committee (ETH2122-1830).

## Analyses

This study used a between-subjects analysis of covariance (ANCOVA) as well as multivariance annalysis of covariance (MANCOVA). The continuous independent variable was dyslexia (Level 1: dyslexia diagnosis, Level 2: no dyslexia). For the ANCOVA, the continuous dependent variable was GIA. For MANCOVA, the continuous dependent variables were SMA and IGD. The study aimed to explore if there was a significant difference between the independent variable and the dependent variables, after controlling for the continuous covariate, age and the nominal covariates, gender, education levels, income levels, and marital status.

## Results

### Descriptive statistics and data screening

Table 2 shows descriptive statistics for all scales. As shown in Table 2, the participants with dyslexia had higher scores than controls on all measures.

A Pearson product-moment correlation was initially run to check for multicollinearity among the dependent variables. While the correlation between GIA and IGD was $r = .61$ and between SMA and IGD was $r = .49$, the correlation between GIA and SMA was $r = .77$. This was deemed to be too high, compared to the acceptable range of *around $r = .8$* for multicollinearity [55]. This suggested that general and specific internet addictions were not sufficiently independent. Hence it was decided that GIA would be isolated for an ANCOVA, while only SMA and IGD would be included in the MANCOVA.

### ANCOVA for GIA

A one-way between subjects ANCOVA was performed to investigate internet-related addictions among persons with and without dyslexia. The dependent variable was IA. The independent variable of interest was dyslexia diagnosis (no dyslexia vs dyslexia diagnosis). The covariates were age, gender, marital status, education, and income levels.

Initial screening of skewness for GIA (skewness = .54; $z = 3.78$) and GIA residuals (skewness = .62; $z = 4.34$) showed positive skewness a significant Shapiro-Wilk (S-W) test ($p <$

**Table 2. Adjusted means and standard deviations of scores.**

| Scale | Dyslexia Group | Controls |
|---|---|---|
| IAT | 40.87 (4.21) | 35.78 (4.36) |
| IGDS-SF9 | 19.82 (2.21) | 16.55 (2.29) |
| BSMAS | 15.41 (1.50) | 14.23 (1.55) |

Standard deviations are presented in parenthesis. IAT = Internet Addiction Test; IGDS-SF9 = Internet Gaming Disorder Scale, Short Form (9); BSMAS = Bergen Social Media Addiction Scale.

.001). Visual inspection of the histograms suggested a moderate positive skew. A square root transformation of IA resulted in an approximately normal distribution of the residuals (skewness = -.02; $z$ = .15) to within the +/- 1.96 range and produced a significant S-W ($p$ = .11) tests and thus indicated normality. Visual inspection of the histogram and Q-Q Plot indicated a normal distribution. The linearity assumption was met. Levene's test of equality of error variance was also satisfactory (p = .69), indicating homogeneity of variances. The adjusted mean GIA score (untransformed) for the no dyslexia and dyslexia groups was 35.78 and 40.87 respectively. After square root transformation, this difference was statistically significant, after controlling for age, gender, income levels, employment, and education levels $F(1, 271)$ = 6.01, $p$ = .02. The partial ETA squared ($\eta^2 p$) was .02, thus a small effect. In terms of demographics, only age was negatively and significantly associated with GIA, untransformed $b$ = -.39, $p <$ .001 with a $\eta^2 p$ of .10 (small effect). For continuous variables like age, this beta is interpreted for every one year-increase in age, GIA scores decrease by .39 units. The other demographics were not significantly associated with GIA.

## MANCOVA for SMA and IGD

A one-way between-subjects MANCOVA was performed to investigate SMA and IGD addictions among persons with and without dyslexia. The dependent variables were SMA and IGD. The independent variable of interest was dyslexia diagnosis (no dyslexia vs dyslexia diagnosis). The covariates were age, gender, marital status, education, and income levels.

The initial screening of SMA's residuals showed moderate positive skewness (skewness = .31; $z$ = 2.16) and significant S-W test ($p$ = .001). Visual inspection of the SMA residuals histogram suggested a slightly positive skew. A square root transformation of the SMA reduced the skewness of the residuals (skewness = .08; $z$ = .53) to within the +/- 1.96 range though the S-W ($p$ = .01) test was still significant. However visual inspection of the histogram and Q-Q plots suggested a normal distribution. The linearity assumption was met. The initial screening of IGD residuals showed moderate positive skewness (skewness = 1.01; $z$ = 7.06) and a significant S-W test ($p <$ .001). Visual inspection of the histogram suggested a moderately positive skew. An inverse transformation of the residuals improved the skewness of the residuals (skewness = -.22; $z$ = 1.55) although the S-W test was still significant (p < .001). The transformed histogram showed a modest negative skew. The linearity assumption was met.

Multivariate outliers and normality were assessed using Mahalanobis distance (MD). Using the untransformed SMA and IGD, there was one multivariate outlier exceeding the critical value of 13.82 for two dependent variables [56]. However, using the appropriately square root transformed SMA and inverse transformed IGD resulted in no multivariate outliers. Homogeneity test was satisfactory; the Levene's Test of Equality of Error Variance was insignificant for the square root SMA (.57) and the inverse IGD (.08). The Box's Test of Equality of Covariance value was also insignificant ($F$ = .98, $p$ = .54), thus suggesting that the observed covariance matrices of the dependent variables are equal across groups.

After controlling for age, gender, income levels, employment, and education levels, there was a statistically significant difference between no dyslexia and dyslexia diagnosis on the combined appropriately transformed dependent variables, $F(2, 270)$ = 5.62, $p <$ .001, Wilk's Lambda = .96. The $\eta^2 p$ was .04. suggesting a small effect. The multivariate model also showed that age, $F(2, 270)$ = 13.58, p < .001, Wilk's Lambda = .91, $\eta^2 p$ = .09, and gender, $F(6, 540)$ = 5.76, $p <$ .001, $\eta^2 p$ = .06, Wilk's Lambda = .88, were statistically significant on the combined appropriately transformed dependent variables.

The adjusted mean SMA (untransformed) for the no dyslexia and dyslexia groups was 14.23 and 15.41 respectively. After square root transformation, this difference was not

statistically significant, $F(1,271) = 3.48$, $p = .06$. The $\eta^2 p$ was .01, thus a small effect. The adjusted mean IGD (untransformed) for the no dyslexia and dyslexia groups was 16.55 and 19.82 respectively. After inverse transformation, this difference was statistically significant, $F(1, 271) = 10.9$, $p < .001$. The $\eta^2 p$ was .04, thus a small effect.

The test between subjects effects also showed that gender was significant for SMA only, $F(3, 271) = 6.03$, $p < .001$, $\eta^2 p = .06$, such that men had significantly lower mean SMA scores than woman (untransformed adjusted means 11.75 and 14.33, respectively). The test between subjects effects also showed that age was negatively and significantly associated with SMA, untransformed beta = -.15, $p < .001$, $\eta^2 p = .08$ and IGD, untransformed beta = -.14, $p < .001$, $\eta^2 p = .03$. All other demographic variables were not significant.

## Interactions

A gender x dyslexia status interaction was included in the ANCOVA for GIA. This interaction was not statistically significant $F(1, 270) = .01$, $p = .94$. An age x dyslexia status interaction was included in the ANCOVA for GIA. Consistent with literature that older individuals have lower scores of GIA [37, 41] the older controls showed lower score for GIA (29.08) relative to the younger controls (37.04). In contrast, the score for older participants did not seem to drop as much (38.95) as compared to younger participants with dyslexia (40.84). However, the statistical trend was not significant for the interaction, $F(1, 270) = 3.41$, $p = .07$. An age x dyslexia status interaction was included in the MANCOVA for SMA and IGD. This interaction was not statistically significant $F(2, 269) = .52$, $p = .60$, Wilk's Lambda = 1.00.

## Discussion

This study aimed to examine if differences exist in General Internet Addiction (GIA), Internet Gaming Disorder (IGD), and Social Media Addiction (SMA) between those with and without dyslexia in a UK population after controlling for age, gender, marital status, employment, and income levels. Findings showed a significant difference for GIA and GD, but no significant difference was found for SMA.

The finding that adults with dyslexia had significantly higher levels of GIA as compared to controls supports the first hypotheses. This finding is also supportive of studies reporting a significant relationship between GIA and other learning disabilities such as ASD [5, 9] and ADHD [8]. The present study can extend this literature by showing that dyslexia in addition to ASD and ADHD is associated with GIA, suggesting that this may be a common factor in learning disabilities.

The second hypothesis was also supported as results showed that participants with dyslexia had significantly higher levels of IGD than controls. Again, this finding is supportive of studies which have shown a correlation between IGD and other learning disabilities such as ASD [9] and ADHD [13]. Hence the results in the present study extend these findings to dyslexia, and again suggest this may be a common factor in learning disabilities.

However, the third hypothesis was not supported by the results. It was expected that those with dyslexia would score significantly lower on SMA than controls, however, although it did not reach significance, participants with dyslexia scored slightly higher than controls on SMA. There are several possible explanations for these findings. It may be that those with dyslexia are effectively employing coping strategies (such as using external resources like search engines for spell checking) when using social media. This may have allowed them to mitigate their deficits in writing and reading and still participate in social media activities meaningfully, such that having a dyslexia diagnosis neither increases nor decreases the risk of SMA relative to controls. Hence the results are supportive of studies hinting at such compensating strategies

adopted by these users [e.g., 34, 35, 57]. Another explanation could be that the types of social media used by the participants in this study is not largely written such as Twitter or Facebook but could be picture or video based such as Instagram, TikTok or YouTube. Indeed, research already shows that those with dyslexia use YouTube as a coping strategy to learn new information [35]. TikTok, in particular, has seen a large rise in usership in recent years, especially amongst adolescents and younger adults [58], and research into this area needs to reflect this change in how we use social media. Future studies could consider if there are differences in the different types of social media used by those with dyslexia.

Given that SMA scores were not significantly higher in the dyslexia group, this suggests that not all learning difficulties are associated with social media addiction. Though ADHD may be correlated with SMA [57] studies show this is not necessarily the case for ASD [15], and the results of this study indicate this may not be the case for dyslexia either. This suggests that, unlike IGD and GIA, SMA might not be a common factor across learning disabilities and instead it could depend on the characteristics of the specific condition. For instance, it is perhaps the language defects seen in ASD including challenges with learning to read [59] and spelling [60] that may limit these individuals' social media usage in a similar way to those dyslexia.

In terms of social demographics, the univariate and multivariate results showed that age was negatively and significantly associated with GIA, SMA, and IGD. This is in line with literature that has suggested that age is significantly correlated with GIA [37], SMA [38], and IGD [38]. In this study, only gender and SMA showed statistical significance, such that the female gender was significantly associated with SMA. This is also in line with previous studies [44].

No interactions were found for gender by dyslexia for GIA, gender by dyslexia for SMA and IGD, age by dyslexia for SMA and IGD. However, age by dyslexia for GIA showed a statistical trend. Consistent with literature that older individuals have lower scores of GIA, the older controls showed lower score for GIA relative to the younger controls. In contrast, the score for older participants did not drop as much as compared to younger participants with dyslexia. This appeared to suggest that age does not moderate GIA levels among those with dyslexia, however the relationship approached but did not reach statistical significance. It is possible that this study was not adequately powered to test for such an interaction effect. Future studies could test this relationship again with larger samples.

Taken together the main findings may suggest that internet addiction is more prevalent in those with dyslexia. This said, it must be noted that although those with dyslexia were found to have higher levels of GIA and IGD this did not fall within pathological levels with group means suggesting only a mild addiction. Therefore, although those with dyslexia might be more likely to show addictive behaviour this is not necessarily a cause for concern. Moreover, it should also be noted that in all three scales, standard deviations show that the two groups are highly overlapping, and differences are only statistically significant after square root transformations therefore suggesting that, although significant, these differences are small.

The current study was preliminary with the aim of exploring if differences exist compared with controls. The findings suggest that this is an area that now warrants further attention. It is possible that the widely reported challenges those with dyslexia face at work, and the accompanying emotional disturbances [5] may be further aggravated by levels of internet addictions or may be pushing them towards higher levels of internet addictions. It is therefore important that future work explores the mechanisms behind these relationships. Additionally, as age is believed to be inversely related to such addictions, it is important for professionals working with younger people who have dyslexia to consider such matters in their assessments and interventions. Future studies could also focus on younger populations to see if the findings extend to adolescents and children. Also, studies could examine more directly the relationships between such addictions and spelling difficulties and information integration.

A key limitation of this study is the cross-sectional nature which precludes conclusions over causality and direction and does not tell us anything about how these relationships operate. One possible explanation for the link between internet addictions and learning disabilities is that learning disabilities may lead to mental health issues, which in turn lead to internet addictions [see 25], or even that mental health mediates the relationship. As this was a preliminary investigation exploring this is beyond the scope of this study and here, to avoid confounding effects, we limited participation only to those who did not have active mental health. This said, it is certainly possible that in our sample, anxiety and depression presented at sub-clinical levels or was undiagnosed. To explore this further, future research may wish to study self-esteem and anxiety (commonly associated with dyslexia; [23]) which may explain a larger amount of variance related to levels of internet addictions or even play a mediating role in the relationship. In doing this the research would be able to understand further how these relationships operate. Additionally, we did not check for the presence of dysgraphia (a writing disability that causes a person's writing to be distorted or incorrect which can be co-morbid with dyslexia; [61]) in our sample. A comorbid diagnosis of dysgraphia could further complicate the relationship between dyslexia and internet-based addictions, in particularly SMA, and this should therefore be explored in future work.

Despite this limitation this study has made a notable contribution to this research area showing that in addition to ASD and ADHD, dyslexia is also related to GIA and IGD. This is important because these findings suggest that internet addictions (at least GIA and IGD) are likely to impact a much larger ambit of people than previously assumed (not just ASD and ADHD). Hence further attention is warranted because if significant relationships between dyslexia and GIA and IGD are detected early, then interventions can be undertaken to manage such problems for this group.

In conclusion, this study was a preliminary investigation into possible differences in terms of GIA, IGD and SMA between those with and without dyslexia in a UK population. Controlling for age, gender, marital status, employment, and income levels, it was found that adults with dyslexia had higher levels of GIA and IGD as compared controls. However, these participants did not show any significant difference in terms of SMA. The results indicate that internet addictions may have a larger ambit for learning disorders beyond just ASD and ADHD and is a hidden problem for users with dyslexia.

## Supporting information

**S1 Data.**
(XLSX)

## Acknowledgments

We thank our study participants for their time.

## Author Contributions

**Conceptualization:** Suresh Kumar, Sophie Jackson, Dominic Petronzi.

**Data curation:** Suresh Kumar.

**Formal analysis:** Suresh Kumar.

**Funding acquisition:** Suresh Kumar.

**Investigation:** Suresh Kumar.

**Methodology:** Suresh Kumar, Sophie Jackson, Dominic Petronzi.

**Project administration:** Suresh Kumar.

**Supervision:** Sophie Jackson, Dominic Petronzi.

**Writing – original draft:** Suresh Kumar.

**Writing – review & editing:** Sophie Jackson, Dominic Petronzi.

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
