## [Decision Letter · Decision Letter 0]

11 Oct 2022

PONE-D-22-21248

A preliminary study into internet related addictions among adults with dyslexia

PLOS ONE

Dear Dr. Sophie Jackson,

Thank you for submitting your manuscript to PLOS ONE. After careful consideration, we feel that it has merit but does not fully meet PLOS ONE’s publication criteria as it currently stands. Therefore, we invite you to submit a revised version of the manuscript that addresses the points raised during the review process.

Please revise your paper. Please follow the reviewers' suggestions below.

We look forward to receiving your revised manuscript.

Kind regards,

Gabriella Vizin, PhD

Academic Editor

PLOS ONE

Journal Requirements:

2. Please change "female” or "male" to "woman” or "man" as appropriate, when used as a noun (see for instance https://apastyle.apa.org/style-grammar-guidelines/bias-free-language/gender).

"The cost of recruiting the participants was sponsored by a donor."

“No potential conflict of interest was reported by the authors.”

Reviewers' comments:

Reviewer's Responses to Questions

**Comments to the Author**

1. Is the manuscript technically sound, and do the data support the conclusions?

Reviewer #1: Yes

Reviewer #2: Yes

2. Has the statistical analysis been performed appropriately and rigorously? 

Reviewer #1: Yes

Reviewer #2: Yes

3. Have the authors made all data underlying the findings in their manuscript fully available?

Reviewer #1: Yes

Reviewer #2: No

4. Is the manuscript presented in an intelligible fashion and written in standard English?

Reviewer #1: Yes

Reviewer #2: Yes

5. Review Comments to the Author

Reviewer #1: The manuscript describes a technically sound piece of scientific research with data that supports almost all conclusions.

The data provided supports almost all conclusions, as noted in the review, data considering age is required from Authors. The manuscript is presented in an intelligible fashion and written in standard English.

Reviewer #2: Review of manuscript PONE-D-22-21248 (“A preliminary study into internet related addictions among adults with dyslexia”) by Sophie Jackson et al.

The authors investigated differences in self-rated general internet addiction (GIA), internet gaming disorder (IGD) and social media addiction (SMA) between a group of participants with and without diagnosis of dyslexia. Results showed that while IGD and GIA were higher in the group with dyslexia compared to the control group, the level of SMA was similar. The authors conclude that internet addiction might be a hidden problem for adults with dyslexia.

The topic of the study is relevant and its novelty is that is that the relationship between internet-related addiction and dyslexia has been an understudied area. The Introduction is well-written and the aims and hypotheses of the study are clearly formed. However, I have some comments and suggestions related especially to the methods and results, which could improve the manuscript readability and understandability in my opinion. I also suggest to check the manuscript in general for typos and inconsistent usage of some words and abbreviations.

My major concerns are the followings:

Introduction:

- Introduction should include a clear definition of dyslexia.

- The authors state that there are diverse results on the relationship between SMA and ASD. Could it be due to the different age groups (and probably different severity of the condition) used in the cited studies (children vs adolescents vs adults), and that different age groups use social media for different purposes? Moreover, it seems to be reasonable that for adults with ASD using written online communication to connect others might be more convenient than for example a phone call or a personal contact.

Methods and results:

- Did the authors check the presence of dysgraphia as well? As persons with dysgraphia might have also serious difficulties with typing in addition to the handwriting, one can hypothesize that this condition is also related to problematic internet and social media usage. Moreover, as authors argue that dyslexia affects writing and spelling skills, the simultaneous presence of dysgraphia (which is quite common) could enhance anxiety when using social media based on writing.

- Page 9, Table 1: how can be the percentage of the widowed/divorced participants 829% of the sample? I think that this might be a typo.

- Page 12: What was the reason that SMA and IGD (r=.49) were submitted into the MANOVA while there was a stronger correlation between GIA and SMA (r=.77)? Does IA and GIA refer to the same construct? If yes, these abbreviations should be consistent.

- Page 13: p = .05 and p = .11 are not significant results of normality tests, suggesting that the distribution of the data met normality.

- Do beta values reflect the differences between groups or do they reflect something else? The authors should clarify.

- There are many inconsistencies in reporting results. When reporting p values, instead of p = .00 authors should report either the exact p value or p < .001. Similarly, authors either use partial ETA square or eta or partial eta in the manuscript. I think that the authors should be more consistent (especially if these expressions are the same), and that would be simpler and more parsimonious to use η2p.

- The authors argue that the lack of predicted effects might be due to the low level of statistical power of the study. Indeed, calculating post-hoc sensitivity analysis could better underpin this statement.

Discussion

- The authors argue that participants with dyslexia might use compensational strategies when using social media. Although some strategies (e.g., spelling and grammar check) are mentioned in the Introduction, it would be helpful to reflect to these strategies again in a more exact way.

- The authors also state that the type of social media (visual such as Instagram or TikTok) or verbal (such as Twitter or Facebook) might influence results. As there is mentioned in the Introduction that persons with dyslexia prefer Youtube videos for learning, I think that the potential role of the dominating type of social media platforms in the null effect should be emphasized more in in the manuscript.

Minor comments

- Page 5: “Google” should be written instead of “Goggle”

- Page 13: authors wrote “sccore” instead of “score”

- The number of decimals is not consistent across the manuscript.

- Interactions would be easier to read in the format e.g., “age x dyslexia status” instead of “age by dyslexia status”.

- I suggest to write “Wilk’s” instead of “Wilk”.

- There should be a space between the two degrees of freedom in ANOVA results.

- Why did the authors apply both Shapiro-Wilk and Kolmogorov-Smirnov tests for normality testing while only one of these should be efficient? Furthermore, the full name of the tests should be marked at the first appearance in the text before using abbreviations.

- Page 15: there is a missing “b” in “lambda”, and there are several unnecessary decimal points and spaces when reporting p values.

6. PLOS authors have the option to publish the peer review history of their article (what does this mean?). If published, this will include your full peer review and any attached files.

Reviewer #1: No

Reviewer #2: No

---

## [Author Response · Author response to Decision Letter 0]

16 Nov 2022

We would like to thank the reviewers for their comments, which have helped improve the manuscript. Please see our categorical responses in red to the comments from both the reviewers. Please note in addition to these changes suggested by the reviewers we have also made some grammar and proof reading amendments. 

Response set 1 

1.0 Reviewer #1: The manuscript describes a technically sound piece of scientific research with data that supports almost all conclusions.

The data provided supports almost all conclusions, as noted in the review, data considering age is required from Authors. The manuscript is presented in an intelligible fashion and written in standard English.

We thank the reviewer for their kind comments. Age is now included see lines 284-285. 

Response set 2 

Introduction

2.1 Page 3: “There are several reasons to suspect that dyslexia might be associated with these types of additions.” The authors should further explain in the manuscript what they mean by several reasons.

This has been edited in order to make it clear that reason the link is likely is because research shows this relationship in other similar populations, please see lines 76-78. 

2.2 Page 4, paragraph 2: The Authors explain how mental health issues as consequences of ASD and ADHD may lead to internet addictions. In order to do so, they line up articles about anxiety, depression and low self-esteem in children with ASD and ADHD, and depression and anxiety as antecedent factors for internet related addictions. Importantly, the Authors base their hypotheses on these associations, as they imply that there is a similar association between dyslexia and internet related addictions. A more thorough explanation of how learning disabilities and internet related addictions might be associated is necessary, especially that the current data focuses on adults and some of the literature is about children. 

We thank the reviewer for this comment and believe the changes we have made in order to address this have strengthened this section of the manuscript. See lines 82 to 94 and 96 to 109.

2.3 Page 4, paragraph 3: A thorough and well written explanation about how SMA and dyslexia might be associated is presented. This would be necessary in the previous paragraph as well. 

We believe that the changes that we have made to address the previous point have also addressed this. Additionally, we have now made some changes including a re-ordering of paragraphs in order to make our arguments clearer (see lines 

183 to 229). 

2.4 Page 6, paragraph 1: “Yet coping strategies may help mitigate the challenges and therefore research is needed to identify if those with dyslexia are susceptible to SMA, in the same way that those with ADHD are.” The Authors do not show literature or research on the comparison between ADHD and dyslexia, thus I suggest to take this comparison out. 

This comparison has been removed. 

2.5 Page 7, present study: Addiction is twice spelled as ‘addition’, please correct.

This has been corrected in lines 65, 67, 77, 85, 94, 242, 252, 253 and 577. 

Methods

2.6 Page 8, participants: Authors state that all participants, including participants with dyslexia have no active mental health issues, however the assumption that dyslexia is relatable to internet addiction lies on the fact that people with dyslexia have higher levels of anxiety and depression. Was this controlled in the Prolific survey platform, and if so, how?

We acknowledge this point, which is a good one. Anxiety and depression may present as comorbid conditions with dyslexia but not always, and for this preliminary paper, to avoid confounding effects, we limited participation only to those who do not have active mental health. This said, it is certainly possible that in our sample anxiety and depression could be presenting at sub-clinical levels or be undiagnosed and therefore serve as partial mediators or moderators. However, as this is a preliminary study this goes beyond the scope of but paper. We do however, discuss this as potential areas for future research in the discussion and this section has been expanded for clarity see lines 579 to 583. 

2.7 Page 9, sociodemographic characteristics of participants: Please provide age of participants as well.

Age is now included in lines 284- 285. 

2.8 Page 9, sociodemographic characteristics of participants: Data is fitted according to marital status, income, education and employment, however gender is not balanced, as male participants are almost double (n=186) compared to female (n=100). If this is a general sociodemographic ratio, it would be important to mention this in the introduction and how it might effect the association between learning disabilities, mental health issues and internet addiction.

We agree with the reviewer. Gender/socio-demographics were already discussed in the introduction. However, this section has been expanded in light of this comment (see lines 239 – 246). Additionally, gender was controlled for in the study to ensure outcomes are not influenced by this 

2.9 Page 11: Suggestion to use ‘Analyses’ instead of ‘Analytical strategies’ as subtitle.

Corrected to ‘Analyses’ (see line 350). 

Results

2.10 Page12, Descriptive statistics and data screening: Descriptive statistics show that both dyslexia and control group fall into the ‘mild’ IA category, and neither group falls into the pathological category in either IGD or SMA. This is problematic, because in later phases of the manuscript, Authors state that dyslexia is related to IGD and IA, however IA is only mild for both groups, and IGD doesn’t reach pathological levels in neither of the two groups.

Although we agree with the reviewer’s sentiment, here we are consistent with the approach in the literature, in that such addictions are not categorical (addicted vs not addicted) but rather that such addictions lie on a dimension/continuum. Hence it is the levels of addictions that are being compared. Thus, for both scales, the higher the score, the higher the addictive behavior. However, in order to acknowledge the reviewer’s point we have added in a caveat to the discussion and toned our conclusion down somewhat (see lines 551-559). 

2.11 The authors imply that the dyslexia group shows higher results in all three scales, however with the standard deviations in mind, the two groups are highly overlapping, differences are only statistically significant after square root transformations, which is explained later. These significant differences don’t imply that participants with dyslexia have IGD. Other than the comment above, results are clearly written and well explained.

We agree with the reviewer’s caution here. In addition to the above caveat, we have added a further caveat which we hope the reviewer feels addresses this point (line 565.) 

2.12 Page 13, line 15: please correct ‘sccore’ to score

Amended in line 387. 

Discussion

2.13 Page 19 paragraph 2: it is not clear from the manuscript what the Authors mean by ‘hidden problem’ particularly for people with learning disabilities. Please explain this a bit more in the introduction and the discussion of the manuscript.

We agree with the reviewer that phrasing was confusing, we have therefore changed it for clarity (see line 551). 

2.14 Page 20, paragraph 2: IGD scores are higher for participants with dyslexia, however concerning the level of scores on the scales, it seems slightly far-fetched to state that it is related to an actual addiction. 

Here we are arguing that there is a statistical difference in terms of levels of IGD between both groups, with the scales suggesting that higher scores are indicative of higher levels of addiction. For clarity on this we have added the word “levels” to line 565. 

2.15 Page 20, paragraph 2: “Hence further attention is warranted because if significant relationships between dyslexia and GIA, SMA and IGD are detected early, then interventions can be undertaken to manage such problems for this group.”. Importantly, this preliminary study SMA was not higher for participants with dyslexia, therefore it is suggested to exclude it from this assumption.

SMA has now been removed in line 599. 

Response set 3

Introduction:

3.1. Introduction should include a clear definition of dyslexia.

We had already included a definition, but we have rewritten the sentence for clarity (see lines 71-74). 

3.2 The authors state that there are diverse results on the relationship between SMA and ASD. Could it be due to the different age groups (and probably different severity of the condition) used in the cited studies (children vs adolescents vs adults), and that different age groups use social media for different purposes? Moreover, it seems to be reasonable that for adults with ASD using written online communication to connect others might be more convenient than for example a phone call or a personal contact

The section on ASD and SMA has now been expanded to address these comments (see lines 104-109). 

Methods and results:

3.3. Did the authors check the presence of dysgraphia as well? As persons with dysgraphia might have also serious difficulties with typing in addition to the handwriting, one can hypothesize that this condition is also related to problematic internet and social media usage. Moreover, as authors argue that dyslexia affects writing and spelling skills, the simultaneous presence of dysgraphia (which is quite common) could enhance anxiety when using social media based on writing.

This was outside of the scope of the current preliminary study. However, as the reviewer states, this certainly warrants future investigation. We have therefore added a discussion of this see lines 587-592. 

3.4. Page 9, Table 1: how can be the percentage of the widowed/divorced participants 829% of the sample? I think that this might be a typo.

This was a typo error it now reads 8 (see Table 1 in line 299).

3.5 Page 12: What was the reason that SMA and IGD (r=.49) were submitted into the MANOVA while there was a stronger correlation between GIA and SMA (r=.77)? Does IA and GIA refer to the same construct? If yes, these abbreviations should be consistent.

Pallant, (2020)’s recommendation is that “correlations up around .8 and .9” are reason for concern and that when this is the case you need to consider removing one variable. Hence, we felt that .77 was approaching .8 and it would be better to isolate GIA from SMA and IGD. We have made this decision clearer in the manuscript see lines 375-379. 

In relation to the abbreviations. Indeed, IA and GIA are the same construct, and this was a consistency error. Changes have now been made to address this in lines 305, 375, 378.

3.6 - Page 13: p = .05 and p = .11 are not significant results of normality tests, suggesting that the distribution of the data met normality.

Here we meant after transformation. We agree with the reviewer that the previous wording was confusing and have therefore edited for clarity (see the paragraph beginning on line 388). 

3.7 Do beta values reflect the differences between groups or do they reflect something else? The authors should clarify.

They reflect between groups; this has now been clarified in lines 403-405. 

 3.8 There are many inconsistencies in reporting results. When reporting p values, instead of p = .00 authors should report either the exact p value or p < .001. 

This has been correct throughout the manuscript.

Similarly, authors either use partial ETA square or eta or partial eta in the manuscript. I think that the authors should be more consistent (especially if these expressions are the same), and that would be simpler and more parsimonious to use η2p.

We agree and have changed to η2p throughout. 

3.9 - The authors argue that the lack of predicted effects might be due to the low level of statistical power of the study. Indeed, calculating post-hoc sensitivity analysis could better underpin this statement.

While post-hoc power analysis could provide exact power, its computation is complex (not estimable with G-power) and beyond the scope of this paper. We believe the reader would accept our argument that a marginally significant p value could become more significant with more participants, which was what we explicitly stated when we wrote in lines 548-549 “Future studies could test this relationship again with larger samples”. 

Discussion

3.10 - The authors argue that participants with dyslexia might use compensational strategies when using social media. Although some strategies (e.g., spelling and grammar check) are mentioned in the Introduction, it would be helpful to reflect to these strategies again in a more exact way.

A reference to this has now been added to the discussion (see lines 514-519). 

3.11 - The authors also state that the type of social media (visual such as Instagram or TikTok) or verbal (such as Twitter or Facebook) might influence results. As there is mentioned in the Introduction that persons with dyslexia prefer YouTube videos for learning, I think that the potential role of the dominating type of social media platforms in the null effect should be emphasized more in in the manuscript.

A reference to this and short discussion has now been added to the discussion (see lines 514-520). 

Minor comments

3.12 - Page 5: “Google” should be written instead of “Goggle”

Corrected in line 206. 

3.13 Page 13: authors wrote “sccore” instead of “score”

Corrected in line 297. 

3.14 The number of decimals is not consistent across the manuscript.

This has now been correct so that we always round to 2 decimal places. 

3.15 Interactions would be easier to read in the format e.g., “age x dyslexia status” instead of “age by dyslexia status”.

This has been corrected in lines 467, 468, 475. 

3.16 - I suggest to write “Wilk’s” instead of “Wilk”.

This has been corrected throughout. 

3.17 - There should be a space between the two degrees of freedom in ANOVA results.

This has been corrected throughout. 

3.18 - Why did the authors apply both Shapiro-Wilk and Kolmogorov-Smirnov tests for normality testing while only one of these should be efficient? Furthermore, the full name of the tests should be marked at the first appearance in the text before using abbreviations.

Shapiro-Wilk test was retained, and the Kolmogorov-Smirnov test was removed. Additionally, the full name of the test was given at the first appearance (see line 389– 394). 

3.19 - Page 15: there is a missing “b” in “lambda”.

Lambda has been corrected in lines 444-445. 

Response set 4 (journal requirements) 

 4.1. Please ensure that your manuscript meets PLOS ONE's style requirements, including those for file naming. The PLOS ONE style templates can be found at

We have made some formatting changes to the manuscript so that formatting is in line with the PLOS ONE style templates. This includes changes to headings and tables. 

2. Please change "female” or "male" to "woman” or "man" as appropriate, when used as a noun (see for instance https://apastyle.apa.org/style-grammar-guidelines/bias-free-language/gender).

These changes have been made throughout. 

"The cost of recruiting the participants was sponsored by a donor."

These have been addressed in the cover letter. 

“No potential conflict of interest was reported by the authors.”

This have been addressed in the cover letter. 

There are no restrictions and therefore we will upload the data set as a supporting information file. 

As outlined above we will now upload the data set as a supporting information file and this is outlined in the cover letter.

---

## [Decision Letter · Decision Letter 1]

4 Jan 2023

A preliminary study into internet related addictions among adults with dyslexia

PONE-D-22-21248R1

Dear Author, 

We’re pleased to inform you that your manuscript has been judged scientifically suitable for publication and will be formally accepted for publication once it meets all outstanding technical requirements.

Kind regards,

Asrat Genet Amnie, MD, EdD, MPH, MBA

Academic Editor

PLOS ONE

---

## [Editor Report · Acceptance letter]

20 Jan 2023

PONE-D-22-21248R1 

A preliminary study into internet related addictions among adults with dyslexia 

Dear Dr. Jackson:

I'm pleased to inform you that your manuscript has been deemed suitable for publication in PLOS ONE. Congratulations! Your manuscript is now with our production department. 

Kind regards, 

on behalf of

Dr. Asrat Genet Amnie 

Academic Editor

PLOS ONE